# Smoking Cessation in Mice Does Not Switch off Persistent Lung Inflammation and Does Not Restore the Expression of HDAC2 and SIRT1

**DOI:** 10.3390/ijms23169104

**Published:** 2022-08-14

**Authors:** Giovanna De Cunto, Simone De Meo, Barbara Bartalesi, Eleonora Cavarra, Giuseppe Lungarella, Monica Lucattelli

**Affiliations:** Department of Molecular and Developmental Medicine, University of Siena, Via Aldo Moro 2, 53100 Siena, Italy

**Keywords:** animal model, cigarette smoke exposure, chronic inflammation, COPD, smoke cessation, disease progression, neutrophil influx, destructive enzymes, oxidative stress, histone deacetylases

## Abstract

Once COPD is established, pulmonary lesions can only progress and smoking cessation by itself is not sufficient to switch off persistent lung inflammation. Similarly, in former-smoker mice, neutrophil inflammation persists and lung lesions undergo progressive deterioration. The molecular mechanisms underlying disease progression and the inefficiency of smoking cessation in quenching neutrophilic inflammation were studied in male C57 Bl/6 mice after 6 months of rest from smoking cessation. As compared with the mice that continued to smoke, the former-smoker mice showed reduced expression of histone deacetylases HDAC2 and SIRT1 and marked expression of p-p38 MAPK and p-Ser10. All these factors are involved in corticosteroid insensitivity and in perpetuating inflammation. Former-smoker mice do show persistent lung neutrophilic influx and a high number of macrophages which account for the intense staining in the alveolar structures of neutrophil elastase and MMP-9 (capable of destroying lung scaffolding) and 8-OHdG (marker of oxidative stress). “Alarmins” released from necrotic cells together with these factors can sustain and perpetuate inflammation after smoking cessation. Several factors and mechanisms all together are involved in sustaining and perpetuating inflammation in former-smoker mice. This study suggests that a better control of COPD in humans may be achieved by precise targeting of the various molecular mechanisms associated with different phenotypes of disease by using a cocktail of drug active toward specific molecules.

## 1. Introduction

Chronic obstructive pulmonary disease (COPD) is characterized by a chronic airflow limitation that is due to a mixture of a small airway disease (bronchiolitis) and alveolar septa destruction (emphysema). Chronic inflammation is the main cause for the destruction of lung parenchyma and narrowing of small bronchioles. Varying from person to person, this condition may be associated with chronic productive cough with mucus hypersecretion, bronchiolar and vascular remodeling, and sometimes the presence of fibrotic areas scattered throughout the parenchyma, in which emphysema and fibrosis may coexist [1].

The pathogenesis of this disease is still subject of research. There is still a poor understanding of the underlying cellular and molecular mechanisms of COPD, which may vary among different phenotypes of this disease [2]. As reported above, COPD is combined with chronic inflammation of airways and lung parenchyma and sometimes with systemic inflammation that may be at the basis of various comorbidities. Although the nature of inflammation in lungs from patients with COPD has been well described, it is still uncertain how this relates to clinical outcomes, disease progression, and response to different therapies. More research is needed to understand this better [3].

The different clinical manifestations that characterize COPD patients suggest the involvement of distinct molecular mechanisms that may be linked to the heterogeneity of inflammatory infiltrate with a prevalence of neutrophils, monocytes/macrophages, and eosinophils, thus, reflecting the secretion of a great variety of proinflammatory mediators. Based on the prevalence of the different blood cells characterizing chronic inflammation, an attempt has been made to associate different inflammatory patterns with COPD phenotypes (inflammatory endotypes). However, these associationa have been clinically not effective since the underlying molecular mechanisms leading to clinical phenotypes are still uncertain. More research is needed to link inflammatory endotypes to clinical manifestations and outcomes in COPD, and to predict responses to precision medicine [4].

Cigarette smoke (CS) is considered to be the main risk factor of COPD in humans, although several other factors such as environmental pollutants, pathogens, or individual genetic susceptibility may play a causal role [5,6]. CS inhalation induces chronic pulmonary inflammatory infiltrates of macrophages, neutrophils, and CD8+ cells that can persist long after smoking cessation [7,8]. In susceptible individuals, this ultimately leads to emphysema characterized by irreversible destruction and dilation of the terminal airspaces of the lung, chronic disability due to respiratory failure, and premature death.

Increased neutrophils in the sputum is a characteristic feature of most COPD patients. The neutrophilic inflammation in COPD is unresponsive to corticosteroids, even at high doses. This may reflect the marked reduction in histone deacetylase-2 (HDAC2) seen in COPD lungs that is secondary to oxidative stress [9,10]. Corticosteroids do not adequately suppress inflammation or reduce exacerbations and disease progression in COPD patients.

The phosphodiesterase-4 inhibitor roflumilast reduces neutrophilic (and eosinophilic) inflammation, and it can reduce exacerbations when added to other treatments. However, the clinical use of this drug is limited by multiple side effects (such as diarrhea, headache, nausea, dizziness, back pain, muscle spasm) that may appear in COPD patients [11].

In recent years, new anti-neutrophil therapies have been investigated but more knowledge is needed on the complex and different mechanisms underlying each clinical phenotype in order to appropriately treat the different inflammatory endotypes. At the present time, smoking cessation (SC) remains to be the most effective therapeutic intervention in patients with COPD [12]. However, in many cases, SC can relieve obstructive symptoms and lessen the inflammatory response but cannot stop progression of the disease [13,14,15,16].

Much of the current knowledge on the pathogenic mechanism(s) implicated in COPD has been derived from studies performed on several animal models and, in particular, on smoking mice, which can replicate several features of human COPD [17,18,19,20]. Several accomplishments in human and in experimental animals have demonstrated that the inflammatory response persists in COPD despite patients having given up smoking. Once the clinical course of COPD is initiated, the pulmonary inflammation continues and the destruction of alveolar airspace and the narrowing of small airways cannot be reversed after SC [15,21,22,23]. The factors and mechanisms underlying the continued influx of inflammatory cells into the lung after removal of the inflammatory stimulus are unknown.

In this study, using an experimental model of COPD induced by chronic CS exposure, we investigated whether changes in the expression of HDAC2 and SIRT1 deacetylases, and in other factors involved in the perpetuation of inflammation and progression of pulmonary lesions, persist after SC. We also investigated whether chronic neutrophilic inflammation, still evident in this model long after SC, is associated with oxidative stress and release of destructive enzymes.

## 2. Results

### 2.1. Effects of Cigarette Smoke on Lung Parenchyma

The lungs of mice exposed to room air for 4 months show a well-fixed parenchyma, normal in appearance (Figure 1A), while the lung parenchyma of mice exposed for 4 months to CS displays foci of mild emphysema disseminated throughout the lung parenchyma (Figure 1B). Morphometric assessment of emphysema by mean linear intercept (Lm) and internal surface area (ISA) shows, at 4 months from the start of the study, mild but significant increases in Lm values, and a significant decrease in the ISA values of smoking mice as compared with the air-control group (Table 1).

### 2.2. Emphysema Lesions Do Progress after Smoking Cessation

At 10 months from the start of the study, the lung parenchyma of the smoking mice shows more severe emphysematous lesions (Figure 1C) than that of the mice exposed to CS for 4 months (Figure 1B). Morphometric evaluation of lung parenchyma at 10 months of smoking exposure confirms the morphological analysis. At this time point, the progression of emphysema lesions is demonstrated by significantly higher Lm values (+7.5%) and lower ISA lung values (−9.0%) as comopared with those observed in smoking mice at 4 months from the start of the experiment (Table 1). No changes in Lm as well as ISA values are found in air-exposed controls at the different time points.

We also analyzed the lung parenchyma of mice exposed to CS for 4 months and then left to rest for further 6 months in room air. This was done to evaluate the evolution of alveolar changes and lung inflammatory state after smoking cessation.

According to what we reported in another study [21], mice allowed to rest for 6 months in room air after 4 months of CS exposure (Figure 1D) display lung changes similar to those present in mice exposed to CS for 10 months (Figure 1C). As can be seen in the slides, evident foci of emphysema disseminated throughout the lung parenchyma are appreciable in both experimental groups (Figure 1C,D). A morphometric evaluation confirms a similar degree of emphysema in mice subjected to smoking for 10 months and those who have stopped smoking for 6 months after 4 months of exposure (Table 1).

The reported findings demonstrate that emphysematous changes continue to progress in smoking mice despite smoking cessation, as if they had never stopped smoking. In this regard, no morphological difference between smoking and former-smoker mice is detected at 10 months from the start of experiment in the total elastin (desmosine) content of the lung (2.64 ± 0.21 vs. 2.75 ± 026 μg/lung, respectively) as well as in its lung morphology and distribution in alveolar septa after Miller’s staining (Figure 1E–G).

### 2.3. Lung Inflammation Does Persist for a Long Time after Smoking Cessation

As is well known, neutrophils and macrophages accumulate in lung tissue after exposure to CS. In our experimental condition, their presence is well appreciable in the lung tissue after 10 months of smoking and in former-smoker mice (Figure 2B,C). Only few macrophages are present in alveolar spaces of controls exposed to air at the same time point (Figure 2A).

A persistent increased number of neutrophils is seen in BALFs of smoking and former-smoker mice as compared with those from mice that have not smoked. Regarding the number of macrophages in the BALFs, we observe a slight initial increase in their number at 4 months of CS exposure (data not shown) and a decrease to almost control levels at 10 months after the start of experiments in smoker and former-smoker mice (Table 2).

This false discrepancy between the histological data (macrophages present in the lung tissue) and those counted in BALFs at 10 months from the start of the experiments is probably due to the different lifespans of the neutrophil (hours) and the macrophage (months), and to a constant influx of neutrophils in the lung and persistence of long-life macrophages in the lung microenvironment.

The continuous influx of neutrophils into the lungs of smoker and former-smoker mice can be sustained by the high levels of cytokine-induced neutrophil-attracting chemokine (KC), a murine analogue of human interleukin-8 (IL-8), and a chemotactic factor for neutrophils.

In lung sections of mice that had smoked for 4 and 10 months, a marked immunohistochemical staining for the chemokine KC is seen in bronchial cells and neutrophils in alveolar septa (Figure 2E). KC is easily observed in the lungs of mice at 6 months from smoking cessation, mostly in inflammatory cells spread throughout the parenchyma (Figure 2F). No reaction for this cytokine is seen in air-exposed mice at the various time points (Figure 2D).

Additional support for the role of KC in sustaining neutrophil inflammation is derived from real-time PCR and ELISA analysis carried out on lung tissues from the various experimental groups at the different exposure times (Figure 2G,H).

The data observed suggest that smoking cessation is not enough to block neutrophilic inflammation in our experimental conditions.

### 2.4. Cigarette Smoke Causes Changes in Histone Deacetylases HDAC2 and Sirtuin-1 (SIRT1)

As widely recognized, oxidants and free radicals derived from cigarette smoke can initiate and sustain lung inflammation in COPD smokers. In addition, CS can induce oxidative modifications of proteins and enzymes damaging their function. In this context, we investigated whether chronic exposure to CS affects the levels of histone deacetylases HDAC2 and SIRT1. These deacetylases were evaluated during the smoking cessation period to understand whether HDAC2 and SIRT1, also known as NAD-dependent deacetylase sirtuin-1, may play roles in the perpetuation of inflammation and progression of the emphysema we observe in mice for a long time after smoking cessation.

The immunohistochemical analysis of HDAC2 and SIRT1 demonstrates that cigarette smoking reduces the expression of these enzymes in lung tissue of smoker mice. On the one hand, in the lungs of air-exposed mice, the HDAC2 enzyme is easily appreciated above all on the bronchial and bronchiolar epithelium of the airways (Figure 3A).

On the other hand, in mice exposed for 4 months to CS, HDAC2 staining appears very weak on all lung structures (Figure 3B), and similarly the reaction appears faint or absent on the lung structures of former-smoker mice (Figure 3C).

After immunohistochemical staining, the SIRT1 enzyme is visible everywhere on the airway epithelium in the lungs of air-exposed animals (Figure 3D). The enzyme shows an appreciable reduction after 4 months (data not shown) or after 10 months of CS exposure (Figure 3E). Similarly, in the lungs of mice that have stopped smoking, SIRT1 reaction appears weak or vanishes (Figure 3F).

### 2.5. Cigarette Smoke Induces p38 MAPK and pSer10 Phosphorylation

It is well known that inflammatory stimuli such as CS can activate p38 MAPK, and subsequently p-p38 kinase is able to phosphorylate serine at position 10 on histone H3 [24,25]. In our series, the immunohistochemical reaction for p-p38 in lung tissue from smoking mice is markedly evident on airway epithelial cells and type II alveolar epithelial cells (Figure 4B). A strong reaction for p-p38 staining remains visible on the bronchi and respiratory epithelium even in the lungs of mice left for 6 months to rest in room air after smoking exposure (Figure 4C). In the lungs of control mice exposed to room air, the re**action f**or this MAP kinase is weak and not noticeable in several lung areas (Figure 4A).

After immunostaining, a positive reaction for pSer10 is seen in lung sections of smoker (Figure 4E) and former-smoker mice (Figure 4F) at 10 months from the start of the study. The staining is particularly evident on sub-bronchial inflammatory cells and peripheral peri-bronchial areas. This reaction is very weak or almost absent in air-exposed mice (Figure 4D).

### 2.6. Neutrophilic Inflammation Is Accompanied by Intense Staining for Neutrophil Elastase, Metalloprotease 9 (MMP-9), and 8-Oxo-7,8-Dihydro-2′-Deoxyguanosine (8-OHdG)

In the lungs of smoker mice, an intense histochemical staining for neutrophil elastase is present on the alveolar septa and within the neutrophils (Figure 5B). A more attenuated but still evident positivity for this enzyme is appreciable on the alveolar septa of former smoker mice at 6 months after smoking cessation (Figure 5C). A negligible positivity for the immunoreaction is present in the lungs of air-control mice (Figure 5A).

In the smoking mice, an immunohistochemical positivity for MMP-9 on airway epithelium and alveolar macrophages is also appreciated (Figure 5E). Unexpectedly, the immunoreaction for MMP-9 is very strong in the airway epithelium and alveolar macrophages even after the cessation of the smoking stimulus (Figure 5F). No positivity is found in air-control animals (Figure 5D).

The persistence of oxidative damage to DNA was evaluated by analyzing the immunohistochemically 8-OHdG in lung slides from smoker and former-smoker mice at 10 months from the start of the study. Evident positivity for 8-OHdG is appreciated on nuclei of parenchymal and bronchiolar cells in the histological slides from the two experimental groups (Figure 5H,I). No positivity for 8-OHdG is found on pulmonary structures from control mice (Figure 5G).

## 3. Discussion

Chronic exposure to cigarette smoke in C57 Bl/6 mice is followed by the onset of pulmonary changes characteristic of COPD in humans. Lung lesions in smoking mice are appreciable from the fourth month of exposure onward and progress over time up to the tenth month from the start of treatment [16,18,21].

Similar to in humans [14], once the disease is established, lung lesions can only progress and smoking cessation by itself is not sufficient to switch off persistent lung and systemic inflammation even in smoking mice.

Lung changes present at 4 months of CS exposure undergo a progressive deterioration over time, so that, after 6 months of rest, we found in former-smoker mice lesions very similar in severity to those found in the group of mice that continued to smoke for a further 6 months. Chronic inflammation in smoking and former-smoking lungs is characterized by lung neutrophilia due to a continuous influx of short-lived neutrophils and by a high number of long-lived macrophages due to their persistence in the alveolar spaces. These cells release a marked amount of serine enzymes or metalloenzymes (neutrophil elastase and MMP-9) able to destroy the lung scaffolding, to activate proinflammatory molecules, or to release large quantities of oxygen radicals that can influence the expression of many proinflammatory mediators. Oxygen radicals induce the redox sensitive transcription factor NF-kB that leads to an increased expression of inflammatory cytokines and chemokines and gives rise to modifications of histone deacetylases. All these factors together with DAMPs released from necrotic cells [21] are involved in sustaining and perpetuating lung inflammation after smoking cessation.

We observe a reduced expression of HDAC2 and SIRT1 both in inflammatory cells and in bronchial and bronchiolar epithelial cells and an increased positivity for the factor KC (homolog of human IL8). The expression of both deacetylases is poor in the lungs of former-smoker mice as well as in smoking mice, suggesting that smoking cessation does not restore the expression of these deacetylases that are likely to have suffered irreversible oxidative damage.

Under our experimental conditions, we also observe a marked expression of p-p38 MAPK and p-Ser10 both in the lungs of smoking and former-smoking mice. This suggests that the p38 MAPK pathway may also be involved in the perpetuation of inflammation, despite the cessation of the stimulus.

It is well known that an imbalance between exogenous oxidants and endogenous antioxidants [7,26] may result in oxidative stress. This process can cause damage of the structural components of lung matrix (such as elastin), injure the airway epithelium, and promote inflammation via upregulation of proinflammatory genes. A major source of oxidants is cigarette smoke (mainly free radicals and nitric oxide) together with inflammatory leukocytes that accumulate in the lungs of smokers and release toxic oxygen radicals. Elevated levels of ROS and their metabolites have been demonstrated in COPD patients and in smoking mice [27,28,29] and the level of oxidative stress was inversely correlated with lung function of the patients [30]. Oxidative imbalance and generation of ROS play important roles in pathophysiology of COPD, which can affect directly or indirectly a variety of proteins, signaling molecules, or receptors [29], thus, altering intracellular and tissue homeostasis.

This relationship is further illustrated by data from animal experiments in which the oxidative stress blockade in transgenic mice overexpressing human Cu/Zn SOD prevented morphological changes in the lungs of COPD models [31], or the targeted deletion of NF-E2-related factor (Nrf2) with the inhibition of the transcription of multiple antioxidant genes caused early-onset and more severe emphysema after chronic cigarette smoke exposure [32,33]. Oxidative stress can participate in the induction of an inflammatory reaction together with other pathogenic factors, and can damage the pulmonary structures (through non-enzymatic modification of macromolecules such as oxidation of α1-antitrypsin and lipids that compose cell membranes, or by favoring the release of neutrophilic elastase, and ny hyperexpression of MMP-9), which can contribute to pathology of COPD through other mechanisms. Certainly, oxidative stress is an important secondary contributor to the disease, more than the primary cause of COPD, causing resistance to corticosteroids [10] and acceleration of normal pulmonary senescence and accumulation of senescent cells [34] in lungs of COPD patients.

The ability of corticosteroids to repress proinflammatory gene expression is impaired by the reduced activity of HDAC2 that is required for inflammatory gene suppression [35]. Oxidative stress in COPD patients induces phosphorylation and ubiquitination of HDAC2 that make it a target for proteosomal degradation and its inactivation by peroxynitrite [36]. At the same time, oxidative stress generated by cigarette smoke causes post-translational modifications of SIRT1. In the lungs of smokers and COPD patients, as well as in bronchial epithelial and monocyte-macrophages cell lines, modifications of SIRT1 (such as nitration of tyrosine residues and carbonylation, formation of 4-hydroxy-2-nonenal adducts with cysteine, lysine, and histidine) have been described [37]. The lack of acetylation of NF-kB caused by the reduced expression of HDAC2 and SIRT1 inevitably causes an increase in the expression of NF-kB dependent proinflammatory genes by both bronchial and alveolar epithelial cells inducing the production of several mediators and cytokines, such as TNF-α and IL-8. These mediators can reinforce the negative effects of a persistent neutrophilia characterizing COPD lungs that “alarmins” (such as formyl peptides) released from mitochondria of necrotic cells sustain [20,38]. In our experimental conditions, oxidative stress (related to the persistent neutrophilia observed in former smoking mice) interferes in at least two signals converging on the NF-kB factor, p-p38 MAPK pathway and the inhibitory pathway from HDAC 2 and SIRT1 deacetylases.

As outline above, oxidative stress contributes in several ways to COPD pathology, and although numerous small molecules evaluated as antioxidants have exhibited therapeutic potential in preclinical studies, clinical trial results have been disappointing [39]. These negative results are based on the negligible effect of scavenging by almost all small molecules, difficulty in achieving effective in vivo concentrations [39], and the declining ability in aging to increase NRF2 activation by NRF2 activators (such as sulforaphane, quercetin, resveratrol, and curcumin) [39,40].

Because of inefficiency of smoking cessation, corticosteroid, and antioxidant therapy, there is an urgent need to achieve a much better control of COPD.

As reported in our study carried out in the mouse strain widely considered to be the human counterpart of the most common COPD phenotype, several factors and molecular mechanisms may be involved all together and simultaneously in the disease pathology and progression. Our study results suggest that better control of COPD may be achieved by specifically targeting the several cellular and molecular mechanisms characterizing the different phenotypes. This will be possible in the future by using a cocktail of drugs active towards specific cells and molecules in the different phenotypes of the disease when pathogenetic mechanisms of the disease are finally elucidated.

## 4. Materials and Methods

### 4.1. Animal Experiments

Male mice from the C57Bl/6 strain (4–6 weeks old) were used in this study, supplied by Charles River Italia (Calco, Italy). The mice were housed in an environment controlled for light (12 am to 12 pm) and temperature (22 ± 2 °C); food (Mucedola Global Diet 2018; Harlan, Corezzana, Italy) and water were provided for consumption ad libitum. All animal experiments were conducted in conformity with the “Guiding Principles for Research Involving Animals and human Beings” and were approved by the Local Ethics Committee of the University of Siena and the Italian Health Ministry (no. 186/2015-PR).

### 4.2. Exposure to Cigarette Smoke

Mice from each experimental group were exposed to either room air (controls) or to the smoke of three cigarettes/day, 5 days/week for 4 or 10 months, (Virginia filter cigarettes, 12 mg of tar and 0.9 mg of nicotine) (smoking group). The methodology for smoke exposure has previously been described in detail [21,41].

A further experimental group included mice exposed to cigarette smoke for 4 months, and then left to rest after cigarette smoke at to room air for the next 6 months.

At the times predetermined in the experimental design, mice exposed to air (15 control mice), to cigarette smoke (15 smoking mice), or at rest after exposure to cigarette smoke (15 former smoking mice) were anesthetized with sodium pentobarbital, sacrificed by severing the abdominal aorta, and the lungs were immediately removed.

### 4.3. Morphology and Morphometry

Lungs were fixed intratracheally with formalin (5%) at a pressure of 20 cm H_2_O. Post-fixation lung volume was measured by water displacement. Lungs were processed for histology and stained with haematoxylin-eosin, and the Miller elastin stain (VWR Chemicals) To improve elastin-to-background contrast, modifications to the Miller’s stain included bypassing the nuclear staining and using a neutral red counterstain in place of the van Gieson counterstain [42].

Lung slides were analyzed for morphology and morphometry. Assessment of emphysema included mean linear intercept (Lm) and internal surface area (ISA) [43,44,45]. For the determination of the Lm and ISA, two blinded pathologists evaluated 40 histological fields for each pair of lungs, both vertically and horizontally.

### 4.4. Inflammatory Cells Profile in BALF

The total and differential cell counts in bronchoalveolar lavage fluids (BALF) were carried out in mice from each experimental group at the end of the treatment. Before removing lungs, the tracheas were isolated in situ in animals under anaesthesia, and then cannulated with a 20 gauge blunt needle. With the aid of a peristaltic pump (P-1 Pharmacia), the lungs were lavaged in situ three times with 0.6 mL saline solution. The average fluid recovery was greater than 95%. The numbers of neutrophils, macrophages, and lymphocytes were assessed by using Diff-Quick staining. Supernatants were used to determine by ELISA the KC content by using a BioPlex immunoassay (Bio-Rad, Hercules, CA, USA), according to the procedure provided by the supplier.

### 4.5. Immunohistochemistry

Lung tissue sections (5 μm thick) were stained for neutrophil elastase (NE), metalloprotease 9 (MMP-9), histone deacetylase 2 (HDAC2), sirtuin-1 (SIRT1), phospho- histone H3 (Ser 10), phospho-p38 (pp38), GRO-alpha (KC), and 8-oxo-7,8-dihydro-2′-deoxyguanosine (8-OHdG).

The primary antibodies (Ab) used were: mouse monoclonal Ab to mouse SIRT1 (1:1000, Invitrogen, ThermoFisher Scientific, Waltham, MA, USA); rabbit polyclonal Ab to mouse HDAC2 (1:20, Invitrogen, ThermoFisher Scientific); mouse monoclonal Ab to mouse phospho-histone H3 (1:100, Invitrogen, ThermoFisher Scientific); rabbit polyclonal Ab to mouse Gro alpha (KC) (1:100, Abcam Ltd., Cambridge, UK); rabbit polyclonal Ab to mouse neutrophilic elastase (NE) (1:100, Abcam Ltd., Cambridge, UK); rabbit polyclonal Ab to mouse MMP-9 (1:200, Novus Biological, Littleton, CO, USA); rabbit polyclonal Ab to mouse pP38 (1:50, Santa Cruz Biotechnology Inc., Europe, Heidelberg, Germany); mouse monoclonal Ab to 8-OHdG (1:500, Abcam Ltd., Cambridge, UK).

To localize mouse primary monoclonal antibodies to SIRT1, phospho-histone H3 and 8-OHdG on mouse tissues, we used the Vector M.O.M. immunodetection kit (Vector Laboratories, Burlingame, CA, USA) containing a novel blocking agent designed specifically to reduce the undesired background staining. Immunostaining was revealed by using the M.O.M. detection kit with 3,3′-diaminobenzidine tetra hydrochloride (DAB) for 8-OHdG; the staining for SIRT1 and phospho-histone H3 was revealed by adding streptavidin-alkaline phosphatase (BD Pharmingen, Buccinasco, Italy).

For antigen retrieval, each tissue section was heated in a microwave for 15 min in citrate buffer 0.01 M, pH 6.0, and allowed to cool slowly to room temperature, then incubated with 3% bovine serum albumin for 30 min at room temperature to block nonspecific antibody binding, and finally incubated overnight at 4 °C with the respective primary antibodies.

To detect HDAC2, the slides were incubated with goat antirabbit IgG- HRP (1:200 Invitrogen) for 40 min, after rinsing PBS containing 0.05% Triton X-100.

To detect Gro-alpha (KC), the slides were incubated with biotinylated goat anti-rabbit IgG (1:100 (Vector Labs, Burlingame, CA, USA) for 40 min, subsequently rinsed with PBS buffer, and then stained using the streptavidin-HRP method (1:1000, BD Pharmingen, Buccinasco, Italy).

To detect MMP-9 and neutrophilic elastase, tissue sections were incubated with goat anti-rabbit IgG (1:200, Sigma, St. Louis, MO, USA) for 30 min at room temperature followed by incubation with peroxidase–anti peroxidase complex, prepared from rabbit serum for 20 min.

For HDAC2, KC, MMP-9, and neutrophilic elastase, the peroxidase reaction was developed with 3,3′-diaminobenzidine tetra hydrochloride (DAB) substrate (Sigma Aldrich, Milan, Italy). As negative controls for the immunostaining, the primary Ab was replaced by non-immunized serum.

To detect pP38 factor (Tyr182), the sections were rinsed and incubated with goat polyclonal anti-rabbit biotinylated IgG (1:100) (Vector Labs, Burlingame, CA, USA) for 40 min at room temperature. The staining was revealed by adding streptavidin-alkaline phosphatase (BD Pharmingen, Buccinasco, Italy).

After rinsing in 0.01 M PBS containing 0.1% Triton X-100, the alkaline phosphatase reaction was developed with NBT/BCIP stock solution (Roche Diagnostics, Milan, Italy) as chromogen diluted in 0.1 M TRIS buffer, pH 9.5, 0.05 M MgCl_2_, 0.1 M NaCl, 2 mM levamisole.

### 4.6. Determination of Desmosine

For the determination of elastin, pieces of fresh lungs were homogenized, processed, and analyzed by ELISA, essentially according to Cocci et al. (2002). Desmosine expressed by μg/lung was taken as an estimate of total lung elastin [46].

### 4.7. RNA Isolation and cDNA Synthesis

Total RNA was extracted from lungs of mice by using Tri Reagent (Ambion, Austin, TX, USA), according to the manufacturer’s instructions. Five mice for each group were used for RNA isolation. RNA was re-suspended in RT-PCR grade water (Ambion), and the amount and purity of RNA were quantified spectrophotometrically by measuring the optical density at 260 and 280 nm. Integrity was checked by agarose gel electrophoresis. Two micrograms of total RNA were treated with TURBO DNase (TURBO DNA-free kit, Ambion) for 30 min and reverse transcribed by using the RETROscript kit (Ambion), according to the manufacturer’s instructions. Two hundredths of the final volume of RT were used for real-time RT-PCR.

### 4.8. Real-Time RT-PCR

Real-time RT-PCR was performed in triplicate for each sample on an MJ Opticon Monitor 2 (MJ Research Co., Waltham, MA, USA) with specific locked nucleic acid probes from the Mouse Universal Probe Library Set (Roche, Indianapolis, IN, USA). Primers were designed by using the free online ProbeFinder software version 2.53 (Roche Molecular Systems Inc., Branchburg, NJ, USA) that shows a pair of specific primers for each probe from the Universal Probe Library set (Roche) (Table 3). The combination of primers and probes provides specific amplification and detection of the target sequence in the sample. PCR reactions were performed in a volume of 25 µL and contained 12.5 µL of FastStart TaqMan Probe Master (Roche), 300 nmol/L forward and reverse primers (TIBMolbiol, Genova, Italy), 200 nmol/L Universal Probe Library Set probes, and 5 µL of cDNA. The real-time RT-PCR assay was performed according to a method reported in detail in our previous studies [19]. Briefly, reactions were incubated at 95 °C for 10 min, and then amplified for 40 cycles, each cycle comprised of an incubation step at 94 °C for 15 s followed by 60 °C for 1 min. The real-time RT-PCR assay included a no-template control and a standard curve of four serial dilution points (in steps of 10-fold) of each of the test cDNAs. The analysis of the results was based on the comparative Ct method (Ct) in which Ct represents the cycle number at which the fluorescent signal, associated with an exponential increase in PCR products, crosses a given threshold. The combination of primers and probes provides specific amplification and detection of the target sequence in the sample (Table 3). The average of the target gene was corrected for 18S rRNA as the endogenous housekeeping gene and normalized to a median control value of 1.0 [47].

### 4.9. Statistical Analysis

Data are presented as mean *±* SD. The significance of the differences was calculated using one-way analysis of variance (F-test). A *p*-value <0.05 was considered to be significant.

## Figures and Tables

**Figure 1 ijms-23-09104-f001:**
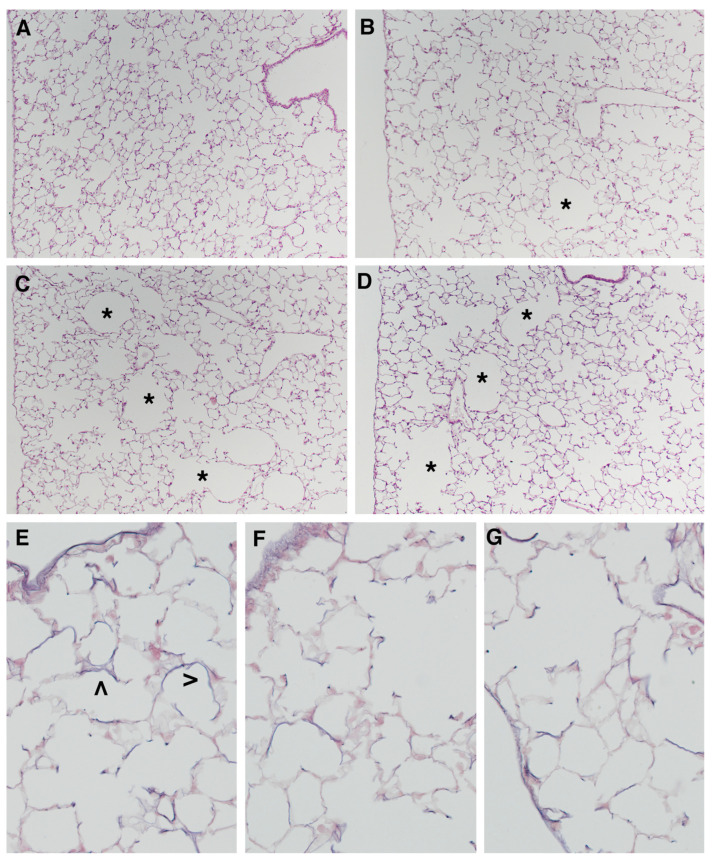
Representative histologic sections from lung parenchyma at different times from the start of the study: (**A**) Lung from an air-exposed mouse showing normal architecture; (**B**) lung from a mouse after 4 months of smoke exposure showing patchy areas of lung emphysema (*); (**C**) lung from a mouse after 10 months of smoke exposure, emphysematous changes are more severe (*) and the confluence of several air spaces due to the loss of the septa is evident; (**D**) lung from a mouse left to rest in room air after 4 months of CS exposure. Note the progression of emphysematous changes in respect to those present in (B); (**E**) elastin (>) distribution in lung from air-exposed mouse, elastin is uniformly distributed normal lung parenchyma; (**F**) elastin distribution in lung from mouse after 10 months of smoke exposure, note the presence of an uneven distribution of the elastin. In some septa elastin fibers appear fragmented or barely visible; (**G**) elastin distribution in lung from mouse left to rest in room air after 4 months of CS exposure. In the periphery of the emphysematous areas, the elastic fibers appear evanescent and fragmented. (**A**–**D**) hematoxylin/eosin stain, ×40 and (**E**–**G**) Miller’s stain for elastin, ×200.

**Figure 2 ijms-23-09104-f002:**
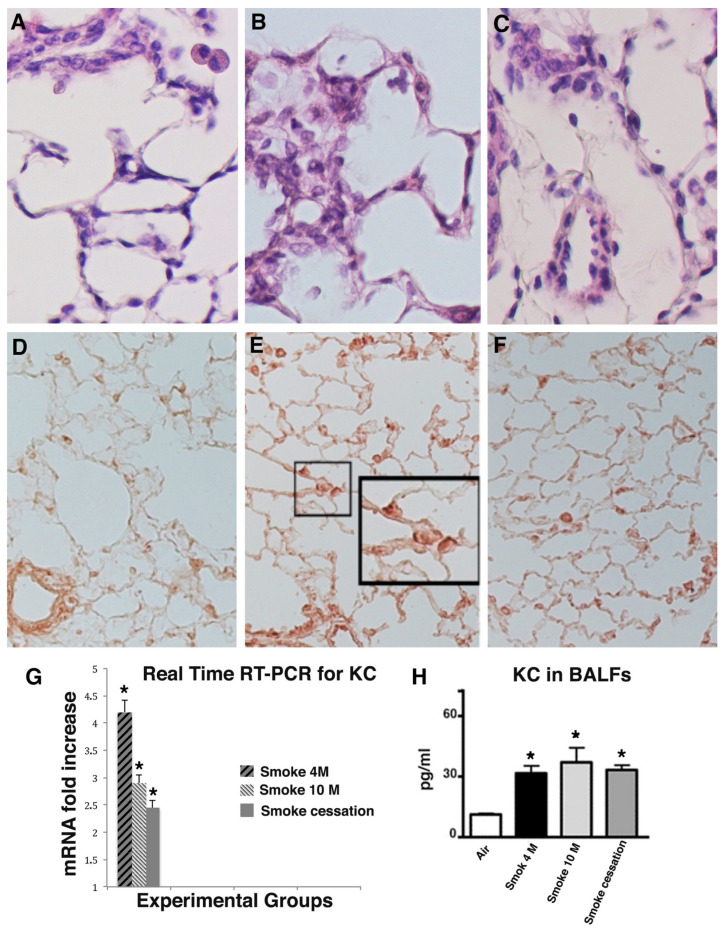
Neutrophilic inflammation and KC expression in lungs of smoking mice: (**A**) Lung from an air control mouse showing very few cells (prevalently macrophages within alveolar spaces); (**B**) lung from a mouse after 10 months of exposure to CS, inflammatory cells are present in air spaces; (**C**) lung from a mouse left to rest in room air after 4 months of CS exposure, a consistent number of neutrophils and macrophages are present in lung parenchyma; (**D**) immunohistochemical analysis reveals a faint reaction for KC in lung parenchyma of air exposed mice; (**E**) in the box high magnification of what contained in the small box. Positive reaction for KC is found in alveolar epithelial cells and macrophages of mice after 10 months of exposure to CS; (**F**) KC can be seen also in alveolar cells and macrophages in lungs from mice left to rest in room air after 4 months of CS exposure; (**G**) real-time PCR analysis of mRNAs for KC carried out on lungs from 6 mice for each experimental group at various time points after the start of experiments. Values of transcripts for KC are corrected for 18S rRNA and normalized to a median control value of 1.0. Error bars indicate mean ± SD. * *p* < 0.05 vs. air controls; (**H**) ELISA analysis of KC in lung tissue and BAL fluids are reported. Error bars indicate mean ± SD. * *p* < 0.05 vs. air controls (**A**–**C**) hematoxylin/eosin stain, ×400 and (**D**–**F**), ×200.

**Figure 3 ijms-23-09104-f003:**
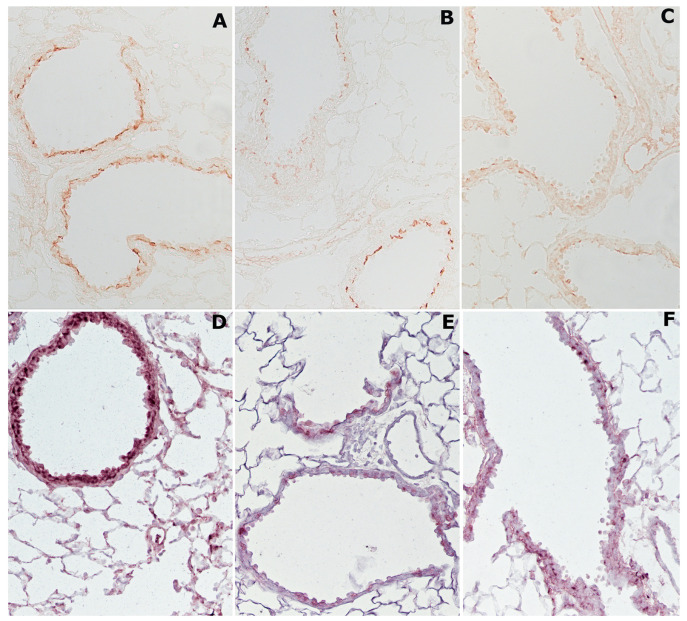
Representative immunohistochemical reaction for histone deacetylases HDAC2 and SIRT1 in mouse lungs from the different experimental groups: (**A**) HDAC2 is usually expressed in lungs from air control mice; (**B**) a marked reduction in the expression of this enzyme is observed in lung tissue of smoking mice at 4 months from the start of the study; (**C**) a similar faint reaction for HDAC2 is observed in lungs from mice left to rest in room air after 4 months of CS exposure; (**D**) SIRT1 enzyme can be seen everywhere on the alveolar epithelium and the airways of air-exposed animals after immunohistochemical staining; (**E**) an appreciable reduction of this enzyme is found in mice at 10 months of CS exposure; (**F**) a weak positivity for SIRT1 is found on sub-bronchial inflammatory cells and peripheral peri-bronchial areas of former-smoker mice. (**A**–**F**), ×200.

**Figure 4 ijms-23-09104-f004:**
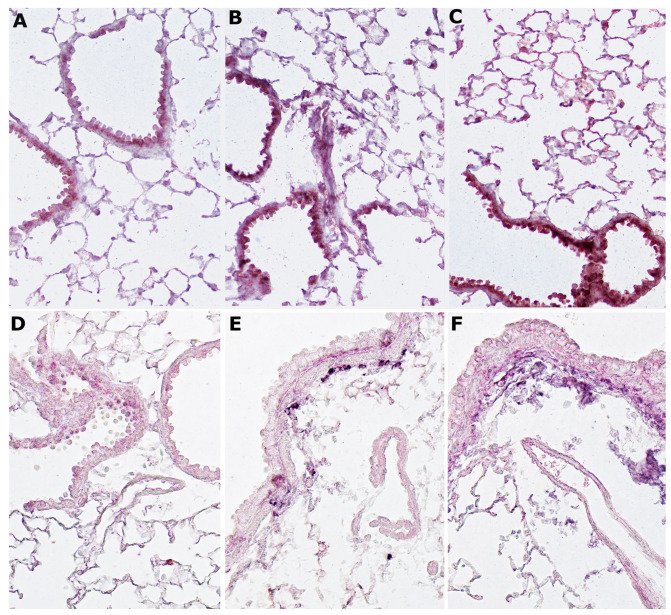
Expression of p38 MAPK and pSer10 phosphorylation after cigarette smoke: (**A**) The immunoreaction for p-p38 is weak and not noticeable in several lung areas of control mice exposed to room air; (**B**) an immunohistochemical positive reaction for p-p38 can be appreciated in lung tissue from smoking mice on airway epithelial cells and type II alveolar cells; (**C**) the reaction for this MAP kinase is also very clear on lung structures of mice that have quit smoking; (**D**) a very weak reaction for pSER10 is present in the lungs of control mice exposed to room air; (**E**,**F**) a strong reaction for pSer10 is present on sub-bronchial inflammatory cells and peripheral peri-bronchial areas of smoking mice (**E**) and in lungs from mice left to rest in room air after 4 months of CS exposure (**F**). (**A**–**F**), ×200.

**Figure 5 ijms-23-09104-f005:**
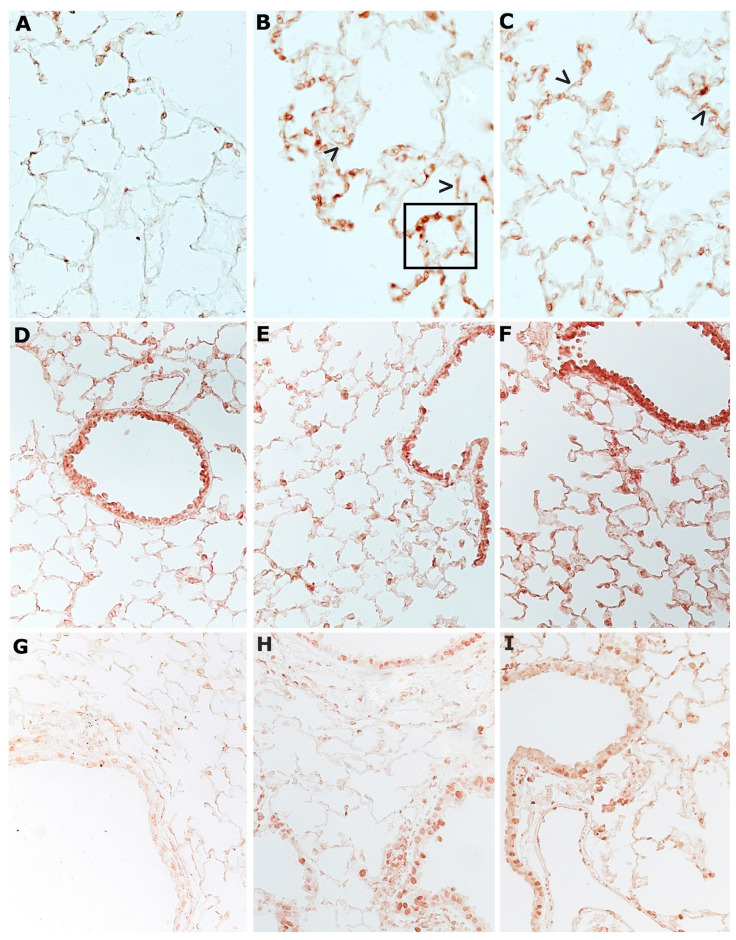
Neutrophil elastase (NE), metalloprotease 9 (MMP-9), and 8-oxo-7,8-dihydro-2′-deoxyguanosine (8-OHdG) after cigarette smoke: (**A**) Immunohistochemical staining for NE in air control lungs is visible only on the few neutrophils present inside the capillaries of the alveolar septa; (**B**) positive staining for NE is seen in the numerous neutrophils that are present throughout the lungs of smoking mice, much of the alveolar interstitium is positive for the immunohistochemical reaction for NE (>); (**C**) NE is seen on alveolar septa (>) and on the numerous neutrophils present inside and outside the alveolar capillaries of lungs from mice left to rest in room air after 4 months of CS exposure; (**D**) immunohistochemical staining for MMP-9, a faint or no positivity for such enzyme if found in alveolar structure of air control mice, on a few cells, probably macrophages, a scant positivity is appreciated; (**E**,**F**) Positivity for MMP-9 on airways epithelium and on alveolar macrophages can be observed in lung from smoking (**E**) and former-smoking mice (**F**); (**G**) immunohistochemical staining for 8-OHdG in lung from air exposed mouse at 12 months from the start of the study showing no positivity; (**H**) positive stain for 8-OHdG is seen on nuclei of parenchymal and bronchiolar cells; (**I**) similar features can be appreciated in pulmonary tissue from former-smoking mice. (**A**–**I**) ×200.

**Table 1 ijms-23-09104-t001:** Lung Morphometry of C56Bl/6J Mice from the Different Experimental Groups.

Experimental Groups	4 Months		10 Months	
	Lm (μm)	ISA (cm^2^)	Lm (μm)	ISA (cm^2^)
**Air**	39.90 ± 1.1	1112 ± 34	40.12 ± 1.1	1020 ± 35
**CS**	42.70 ± 1.1 *	1020 ± 45 *	45.90 ± 2.0 *†	928 ± 48 *†
**CS + SC**			44.98 ± 2.1 *†	947 ± 33 *†

Definition of abbreviations: Air, exposure to room air; CS cigarette smoke exposure; SC, smoking cessation. Lm, mean linear intercept; ISA, Internal surface area of lung. Data are given as mean ± SD from 6 mice/group. * *p* < 0.05 vs. air-exposed mice at the same time point; † *p* < 0.05 vs. CS exposed mice at 4 months.

**Table 2 ijms-23-09104-t002:** Total and Differential Cell Count in BALFs after 10 Months from the Start of the Study.

	Air-Exposed	Smoke-Exposed	Smoke Cessation
**Total cell count, ** **×** **10^5^**	**1.34 ± 0.3**	**1.87 ± 0.42 ***	**1.55 ± 0.16**
**Differential cell count, ** **×** **10^5^**			
**Macrophages**	**1.12 ± 0.2**	**1.29 ± 0.39**	**1.13 ± 0.11**
**Neutrophils**	**0.15 ± 0.06**	**0.47 ± 0.12 ***	**0.33 ± 0.04 *#**
**Lymphocytes**	**0.07 ± 0.03**	**0.11 ± 0.03**	**0.09 ± 0.01 #**

Definition of abbreviations: Data are given as mean ± SD from 5 mice/group. They represent data from mice exposed for 10 months to air, cigarette smoke or mice left to rest for 6 months to room air after smoke cessation. The slides were stained with Diff Quick©. * *p* < 0.05 vs. group exposed to air; # *p* < 0.05 vs. group exposed to cigarette smoke.

**Table 3 ijms-23-09104-t003:** Primers Sequence and Probe Catalog Number.

	Primer Sequence	Probe	Amplicon Length (nt)
KC (CXCL1)	Fw: 5’-AGACTCCAGCCACACTCCAA-3’ Rev: TGACAGCGCAGCTCATTG-3’	#1	86
rRNA 18S	FW: 5’-AAATCAGTTATGGTTCCTTTGGTC-3’ Rev: 5’-GCTCTAGAATTACCACAGTTATCCAA-3’	#55	

## Data Availability

Not applicable.

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
