# Peer review of "Smoking Cessation in Mice Does Not Switch off Persistent Lung Inflammation and Does Not Restore the Expression of HDAC2 and SIRT1"

_ijms, 2022, doi:10.3390/ijms23169104_

Round 1
Reviewer 1 Report
I would like to congratulate the authors for their interesting and informative paper.
This is an animal study investigating the effect of smoking cessation on persistent lung inflammation in male mice of C57Bl/6 strain. Particularly, the authors studied whether changes in the expression of HDAC-2 and SIRT-1 histone deacetylases persist after smoking cessation, as well as whether chronic neutrophilic inflammation, which remains evident long after smoking cessation, is associated with oxidative stress and release of destructive enzymes. They found that former -smoker mice showed reduced expression of HDAC-2 and SIRT-1, as well as marked expression of p-p38 MAPK and p-Ser10, as in the mice that continued to smoke. Furthermore, former-smoking mice demonstrated persistent lung neutrophilic influx and high number of macrophages. Therefore, the authors conclude that better control of COPD in humans may be achieved by targeting precise molecular mechanisms in the different phenotypes of the disease by using a combination of drugs that are active towards specific molecules involved in the pathogenesis of COPD.
This is an overall well written manuscript. Here, I have made a few suggestions that (in my opinion) could help improve the overall quality of the article.
· The authors may consider explaining how the sample size was determined and provide details of any a priori sample size calculation, if performed.
· The authors may consider describing any criteria used for including or excluding mice during the experiment and data points during the analysis. If no criteria were set, they may consider stating this explicitly.
· The authors may consider stating whether randomisation was used to allocate mice to control and treatment groups.
· The authors may consider reporting any blinding at the different stages of the experiment.
· The authors may consider reporting the weight of the mice if such data are available.
· The authors may consider providing further relevant information on the health and immune status of the included mice.
· The authors may consider providing the relevant licence or protocol number of approval by the ethical review committee.
· The authors may consider reporting adverse events, if any, during the experiment.
Author Response
Response (R) to the co suggestions (C) of the Reviewer #1
Suggestions for Authors
(C) “This is an animal study investigating the effect of smoking cessation on persistent lung inflammation in male mice of C57Bl/6 strain. Particularly, the authors studied whether changes in the expression of HDAC-2 and SIRT-1 histone deacetylases persist after smoking cessation, as well as whether chronic neutrophilic inflammation, which remains evident long after smoking cessation, is associated with oxidative stress and release of destructive enzymes. They found that former -smoker mice showed reduced expression of HDAC-2 and SIRT-1, as well as marked expression of p-p38 MAPK and p-Ser10, as in the mice that continued to smoke. Furthermore, former-smoking mice demonstrated persistent lung neutrophilic influx and high number of macrophages. Therefore, the authors conclude that better control of COPD in humans may be achieved by targeting precise molecular mechanisms in the different phenotypes of the disease by using a combination of drugs that are active towards specific molecules involved in the pathogenesis of COPD. This is an overall well written manuscript. Here, I have made a few suggestions that (in my opinion) could help improve the overall quality of the article”.
R. We thank the reviewer #1 for his suggestions and the words of appreciation for our study.
(C) “The authors may consider explaining how the sample size was determined and provide details of any a priori sample size calculation, if performed”.
(R) The sample size was determined based on the previous study carried out by us on smoke cessation in C57 Bl/6 mice (Am J Pathol, 2018:188:2195-2206) and on the experience gained over the last 20 years on the many dozen studies carried out on this animal model (and published) by our group.
(C) “The authors may consider describing any criteria used for including or excluding mice during the experiment and data points during the analysis. If no criteria were set, they may consider stating this explicitly”.
(R) No criteria for excluding or including mice during experiment or data points during the analysis were used, because no inclusion or exclusion were made in this study. As shown in previous studies, the mild lung damage caused by cigarette smoking in the strains does not cause suffering or obvious clinical symptoms such that they should be excluded from the study (Bartalesi B, Cavarra E, Fineschi S, Lucattelli M, Lunghi B, Martorana PA, Lungarella G. Different lung responses to cigarette smoke in two strains of mice sensitive to oxidants. Eur Respir J. 2005; 25:15-22; De Cunto G, Cardini S, Cirino G, Geppetti P, Lungarella G, Lucattelli M. Pulmonary hypertension in smoking mice over-expressing protease-activated receptor-2. Eur Respir J. 2011; 37: 823-834.). The lack of weight loss and the absence of signs of behavioural abnormalities or suffering underline the absence of adverse effects.
(C) “The authors may consider stating whether randomisation was used to allocate mice to control and treatment groups. · The authors may consider reporting any blinding at the different stages of the experiment”.
(R) Randomization is a general procedure in scientific studies. In our opinion is pleonastic include in the manuscript that this procedure was adopted to allocate mice in the different experimental groups. Additionally, as reported in Material and Methods section morphometry was carried out in a blinded manner by two different pathologists. Since male C57 Bl/6 mice, as widely accepted, do develop mild emphysematous lesions, we believed unnecessary to acquire data on the health status of mice we used in this study.
(C) “The authors may consider reporting the weight of the mice if such data are available. · The authors may consider providing further relevant information on the health and immune status of the included mice”
(R) Since male C57 Bl/6 mice, as widely accepted, do develop mild emphysematous lesions, we believed unnecessary to acquire data on the health status of mice we used in this study.
(C) “The authors may consider providing the relevant licence or protocol number of approval by the ethical review committee”.
(R) The protocol number of the approval (no. 186/2015-PR) has been added in the revised manuscript.
Reviewer 2 Report
The manuscript by De Cunto and colleagues shows the ineffectiveness of smoking cessation in the progression of the inflammatory and oxidative process activated by smoking itself in the lungs of a mouse model. The authors correlate the morphological and molecular changes they observe in the lungs of smoke-exposed mice with those that occur in patients with COPD. The work is interesting and addresses a topic for which there is a medical need.
Comments
-Mice exposed for 4 months to smoke and then exposed to air for 6 months are named in different ways throughout the manuscript: former smokers, ex-smokers, etc... I recommend using only one way.
-Some acronyms are defined in full in materials and methods, but the materials and methods section is at the end of the manuscript. Please, define the acronyms the first time they appear in the text.
-In Figure 2, panels G and H require revision: in panel G (RT-PCR), in the legend, months is denoted by mo while in the x-axis of panel H it is denoted by M. Similarly, “cessassion” (to be changed with cessation) in panel G is given in full and in the x-axis of panel H by C. Please, conform the titles of the legends and axes.
-There is no title in the x-axis of panel G, figure 2: please correct.
-Relative mRNA in the G-panel axis of figure 2 is incorrect: amplified transcript should be identified. In materials and methods need to be reported how that transcript was quantified.
-line 101: LM should be changed to Lm.
-Discussion is too long: there are repeated and overlapping concepts from the introduction that require revision.
-Both in the abstract and in the discussion, the conclusions suggested by the experiments are unclear: given that smoking cessation does not slow the progression of the inflammatory and oxidative process, what are the new targets that the authors suggest?
Author Response
Response (R) to the comments (C) of the Reviewer #2
General Comment:
(C) “The manuscript by De Cunto and colleagues shows the ineffectiveness of smoking cessation in the progression of the inflammatory and oxidative process activated by smoking itself in the lungs of a mouse model. The authors correlate the morphological and molecular changes they observe in the lungs of smoke-exposed mice with those that occur in patients with COPD. The work is interesting and addresses a topic for which there is a medical need”.
(R) We thank the reviewer for his words of appreciation for our study.
Specific comments
(C) “Mice exposed for 4 months to smoke and then exposed to air for 6 months are named in different ways throughout the manuscript: former smokers, ex-smokers, etc... I recommend using only one way”.
(R) According to the referee’s suggestion, only the term "former-smoker" has been used throughout the manuscript to define mice exposed for 4 months to smoke and then exposed to air for 6 months.
(C) “Some acronyms are defined in full in materials and methods, but the materials and methods section is at the end of the manuscript. Please, define the acronyms the first time they appear in the text”.
(R) The text of the revised manuscript has been modified accordingly.
(C) “In Figure 2, panels G and H require revision: in panel G (RT-PCR), in the legend, months is denoted by mo while in the x-axis of panel H it is denoted by M. Similarly, “cessassion” (to be changed with cessation) in panel G is given in full and in the x-axis of panel H by C. Please, conform the titles of the legends and axes.”
(R) According to the suggestion, panels G and H have been revised and the titles of legends and axes have been made uniform.
(C) “There is no title in the x-axis of panel G, figure 2: please correct”.
(R) A title in x-axis of panel has been added.
(C) “Relative mRNA in the G-panel axis of figure 2 is incorrect: amplified transcript should be identified. In materials and methods need to be reported how that transcript was quantified”.
(R) G panel has been corrected in the revised manuscript according to the referee’s suggestion and details of the quantification of the transcript has been provided in the Revised version of the manuscript.
(C) ” line 101: LM should be changed to Lm”.
(R) The text has been modified accordingly.
(C) “Discussion is too long: there are repeated and overlapping concepts from the introduction that require revision”.
(R) The discussion section has been shortened accordingly, and the text has been modified to avoid repeated and overlapping concepts from the introduction.
(C) “Both in the abstract and in the discussion, the conclusions suggested by the experiments are unclear: given that smoking cessation does not slow the progression of the inflammatory and oxidative process, what are the new targets that the authors suggest?”
(R) In agreement to the referee comment, Abstract and Discussion section has been modified to render clearer our suggestions.
We thank the reviewer for his constructive suggestions
Reviewer 3 Report
The article “Smoking cessation in mice does not switch off persistent lung inflammation and does not restore the expression of HDAC-2 and SIRT-1” by de Cunto et al. shows that smoking cessation in smoking mice does not reverse the smoke-induced changes in the lungs. This is an interesting finding as COPD is believed to be an “one-way disease” where the pathological changes induced by inflammation cannot be reversed but only stopped or slowed down. There is one thing that makes the article difficult to read; the form of the text. Please consider not to write every sentence in a separate paragraph, to write the ideas rather as a flowing text. In line 48-50 the authors write about different inflammatory endotypes however without enumerating or describing them. Maybe it would be an idea to add some information, some descriptions of the most common types and maybe some extra difficulties in treating them. Please write the sentence about limitation of drugs in line 70-71 in a more clear way. Moreover, pay attention to the formatting of the text (missing citations, not the correct style of citations in the text, different size and style of letters within one sentence etc.), especially in legends of the figures and tables. They should have the same formatting like the text itself. Correct the legends of the tables to make them more legible.
Author Response
Response (R) to the comments and suggestions (C) of the Reviewer #3
Comments and Suggestions
(C) “The article “Smoking cessation in mice does not switch off persistent lung inflammation and does not restore the expression of HDAC-2 and SIRT-1” by de Cunto et al. shows that smoking cessation in smoking mice does not reverse the smoke-induced changes in the lungs. This is an interesting finding as COPD is believed to be an “one-way disease” where the pathological changes induced by inflammation cannot be reversed but only stopped or slowed down”.
(R) We thank the reviewer for his words of appreciation for our study
(C) “There is one thing that makes the article difficult to read; the form of the text. Please consider not to write every sentence in a separate paragraph, to write the ideas rather as a flowing text. In line 48-50 the authors write about different inflammatory endotypes however without enumerating or describing them. Maybe it would be an idea to add some information, some descriptions of the most common types and maybe some extra difficulties in treating them.
(R) The text of the revised manuscript has been modified according to the referee’s suggestion. In addition, more information on the "inflammatory endotypes" of COPD and their current clinical value is now provided in the revised text.
(C) “Please write the sentence about limitation of drugs in line 70-71 in a more clear way”.
(R) A more complete list of side-effects of roflumilast is given in the revised manuscript.
(C) “Moreover, pay attention to the formatting of the text (missing citations, not the correct style of citations in the text, different size and style of letters within one sentence etc.), especially in legends of the figures and tables. They should have the same formatting like the text itself. Correct the legends of the tables to make them more legible”.
(R) As suggested, attention has been paid in formatting the text, and in correcting the legends to make them more legible.
We thank the reviewer for his suggestions.